# Multi-ancestry meta-analysis of genome-wide association studies discovers 67 new loci associated with chronic back pain

Ian B. Stanaway [1,2], Pradeep Suri[1,3,4,5], Niloofar Afari[6,7,8], Daniel Dochtermann [9], Armand Gerstenberger[1,10], Saiju Pyarajan [9], Eric J. Roseen[11,12], Million Veteran Program* & Marianna Gasperi [1,6,7,8,10,13]

This multi-ancestry meta-analysis of genome-wide association studies (GWAS) investigated the genetic factors underlying chronic back pain (CBP) in a sample from the Million Veteran Program comprised of 553,601 Veterans of African (19.2%), European (72.6%), and Hispanic (8.2%) ancestry. The results revealed novel ($N = 67$) and known ($N = 20$) genome-wide significant loci associated with CBP, with 43 independent variants replicating in a non-overlapping contemporary meta-GWAS of the spinal pain dorsalgia phenotype. The most significant novel variant was rs12533005 (chr7:114416000, $p = 1.61 \times 10^{-20}$, OR = 0.96 (95% CI: 0.95–0.97), EA = C, EAF = 0.39), in an intron of the *FOXP2* gene. In silico functional characterization revealed enrichment in brain and pituitary tissues. Mendelian randomization analysis of 62 variants for CBP-MVP revealed 48 with causal links to dorsalgia. Notably, four genes (*INPP5B, DRD2, HTT, SLC30A6*) associated with these variants are targets of existing drugs. Our findings more than double the number of previously reported genetic predictors across all spinal pain phenotypes.

Back pain is the leading contributor to years lived with disability globally[1–3] and is the 4th most common reason for disability in Veterans[4]. The public health burden of back pain is driven mainly by chronic back pain (CBP), which is defined as back pain that persists for ≥3 months[5]. Military Veterans are particularly susceptible to CBP due to physical and psychological stressors experienced during military service, making this a pressing public health concern[6,7].

The etiology of CBP is multifaceted, involving both environmental and genetic elements. Recent research suggests that genetic factors play a crucial role in the development and persistence of CBP[8–10], revealing a polygenic architecture where multiple genetic variations cumulatively contribute to overall CBP risk[8–10]. However, our understanding remains incomplete, especially regarding genetic risk factors across diverse populations.

Differences in CBP prevalence and experience have been observed across gender and racial and ethnic groups. Women are generally at higher risk for chronic pain conditions, including CBP, potentially due to endocrine, psychosocial, and pain perception

[1]VA Puget Sound Health Care System (VAPSHCS), Seattle, WA, USA. [2]Department of Nephrology, University of Washington, Seattle, WA, USA. [3]Seattle Epidemiologic Research and Information Center, VAPSCHS, Seattle, WA, USA. [4]Department of Rehabilitation Medicine, University of Washington, Seattle, WA, USA. [5]Clinical Learning, Evidence, and Research (CLEAR) Center, University of Washington, Seattle, WA, USA. [6]Department of Psychiatry, University of California San Diego, La Jolla, CA, USA. [7]VA San Diego Healthcare System (VASDHS), San Diego, CA, USA. [8]Center of Excellence for Stress and Mental Health, VASDHS, San Diego, CA, USA. [9]Center for Data and Computational Sciences (C-DACS), VA Boston Healthcare System (VABHS), Boston, MA, USA. [10]Mental Illness Research Education and Clinical Center (MIRECC), VAPSHCS, Seattle, WA, USA. [11]Section of General Internal Medicine, Department of Medicine, Boston University Chobanian & Avedision School of Medicine and Boston Medical Center, Boston, MA, USA. [12]Department of Physical Medicine & Rehabilitation, VA Boston Healthcare System, Boston, MA, USA. [13]Department of Psychiatry and Behavioral Sciences, University of Washington, Seattle, WA, USA. *A list of authors and their affiliations appears at the end of the paper. ✉e-mail: mgasperi@uw.edu

differences[11,12]. Furthermore, African American and Hispanic individuals experience higher pain severity and disability compared to non-Hispanic White individuals[13]. These disparities likely result from a combination of genetic susceptibility and environmental influences, such as socioeconomic factors, healthcare access, and cultural differences in pain management[14]. Investigating these differences is essential for understanding the etiology of CBP and developing effective, personalized interventions.

The Million Veteran Program (MVP) provides a unique opportunity to investigate the genetic basis of CBP. This comprehensive bio-bank includes genetic, health, lifestyle, and electronic health record (EHR) data from a diverse cohort of American Veterans, enabling analyses inclusive of various ancestral backgrounds[15].

Given this backdrop, our study was designed with two over-arching aims. First, we aimed to expand knowledge on the genetic basis of CBP by conducting a multi-ancestry Genome-Wide Association Study (GWAS) in military Veteran participants of the MVP. Second, we aimed to examine the extent to which risk factors for CBP are shared across or specific to different ancestral backgrounds. These results could illuminate the genetic landscape of CBP and pave the way for personalized treatments and novel drug targets for chronic pain.

## Results

### Chronic back pain GWAS and functional integration

The study sample included 553,601 participants (Table 1), with 294,723 cases and 258,878 controls from three ancestral backgrounds, resulting in an overall phenotypic prevalence of 53.2%. European ancestry was the largest ancestry group, representing 72.6% of participants (402,005 participants), followed by African American (19.2%) and Hispanic (8.2%) ancestries. Most participants (82.1%) were over 50 years old, with 454,472 individuals in this age group. The average age of the cohort was 61.7 years (SD 14.1), with an average age of 61.2 years (SD 14.2) for European, 57.0 years (SD 12.1) for African American, and 54.1 years (SD 15.5) for Hispanic participants. The gender distribution was predominantly male, 91% ($n = 504,527$) male and 9% ($n = 49,074$) female, consistent with the distribution (10% female) in the U.S. VA population[16]. A significant difference ($p < 0.00001$) with respect to self-reported gender was observed between cases where, in Europeans, 64.6% of women and 53.0% of men had CBP-MVP. There was a higher proportion of women with CBP-MVP as compared to men across all ancestry groups, as seen in Supplementary Data 1. Details on the cohorts and phenotypes are provided in the supplement.

CBP prevalence varied across HARE race and ethnicity groups, with the highest sample prevalence of 62.3% in AFR individuals, 59.6% in individuals with HIS, and 50.1% in EUR individuals.

### Significant GWAS variants

Supplementary Figs. S1–S3 provide the ancestry and ethnicity specific Manhattan plots and Supplementary Data 2–4 have complete lists of the significant variants. In the discovery GWAS of CBP-MVP conducted in participants of EUR ancestry, we identified 2896 genetic variants attaining genome-wide significance, at 68 loci. Among these, 74 SNPs were independent signals by LD ($r^2 < 0.1$). There were five significant variants at two loci in the African ancestry stratum and none in the

Hispanic ethnicity stratum. Multi-ancestry meta-analysis identified 3447 variants attaining genome-wide significance, at 87 loci. Among these, 90 variants were independent signals by LD ($r^2 < 0.1$). None of the lead variants had indication of heterogeneity between ancestries, with only one having an FDR < 1 (0.92). Furthermore, in the genome-wide meta-analysis, there was little evidence of heterogeneity across all variants, with only 52 variants having an FDR < 0.1. Figure 1 shows the Manhattan plot of the meta-analysis.

### Polygenicity

Supplementary Data 5 provides the LDSR summary statistics and Supplementary Figs. S4–S7 show all quantile-quantile (QQ) plots. The Lambda for the EUR stratum was 1.48, and correction by LDSR showed the intercept to be 1.14, indicating polygenicity remained after correction for LD. Lambda for the AFR stratum was 1.07 with an LDSR intercept of 1.03. Analysis of the HIS stratum produced a Lambda of 1.08 with an LDSR intercept of 1.04.

### MVP-CBP genome-wide significant variant replication

In the multi-ancestry meta-analysis of CBP-MVP, 57 variants replicated at the nominal threshold of significance ($p < 0.05$) among 85 independent lead variants which had available summary statistics in Bjornsdottir et al.[17] 26 variants replicated at the Bonferroni-corrected threshold ($p < 0.05/85 = 0.000556$). In the EUR stratum, 49 of the 68 independent variants replicated at the nominal threshold of significance and 23 variants replicated at the Bonferroni-corrected threshold ($p < 0.05/68 = 0.00074$). In the AFR ancestry stratum, 1 of the 2 lead variants replicated at the nominal threshold of significance (rs140875296, $p = 0.046$), where it was reported at MAF = 0.01. rs140875296 is likely AFR-specific (MVP MAF = 0.042, 1000 Genomes African MAF = 0.015) as it is not present in the MVP EUR stratum at >1% MAF nor is it detected in the 1000 Genomes EUR (MAF = 0). rs10119541 was present in the sample reported by Bjornsdottir et al (MAF = 0.037) and does not replicate ($p = 0.76$); while present in both the MVP EUR and HIS strata (MAF = 0.03) but does not replicate ($p = 0.059$, $p = 0.63$ respectively). See Supplementary Data 6–10 for variant replication summary statistics.

### Novel and known variant discovery

To date, ten prior GWAS of back pain-related phenotypes have been published. Table 2 summarizes the counts of loci reaching genome-wide significance in each prior publication. Supplementary Data 11 specifies each publication's reported lead variants and nearby genes and the sheet Supplementary Data 12 tabulates these with findings from the current multi-ancestry meta-GWAS of CBP-MVP. Taken together, prior GWAS of back pain-related phenotypes reported 94 unique lead variants (among 104 lead variants at 100 loci from the 10 publications) which represent 59 distinct risk loci. Eighty-six of the 94 unique lead variants were present in the current GWAS of CBP-MVP conducted in the MVP biobank with the same rsID. Of these 86 variants, 67 replicate at the nominally significant threshold ($p < 0.05$) in the current GWAS of MVP-CBP, and 54 replicate at the Bonferroni-corrected threshold (Supplementary Data 12. The current GWAS conducted in MVP has 87 risk loci that are distinct with one or more lead

**Table 1 | CBP-MVP study samples in the discovery GWAS by HARE-based ancestry and ethnicity**

|  | All Ancestry and Ethnicity | | | European | | | Hispanic | | | African | | |
|---|---|---|---|---|---|---|---|---|---|---|---|---|
|  | Total | Case | Controls | Total | Case | Controls | Total | Case | Controls | Total | Case | Controls |
| N (%) | 553,601 | 294,723 (53.2) | 258,878 (46.8) | 402,005 (78.7) | 201,443 (50.1) | 200,562 (49.9) | 45,131 (8.8) | 26,914 (59.6) | 18,217 (40.4) | 106,465 (20.8) | 66,366 (62.3) | 40,099 (37.7) |
| Age (SD) | 61.7 (14.1) | 59.6 (13.8) | 63.9 (14.2) | 63.7 (13.8) | 61.5 (13.7) | 65.9 (13.5) | 54.6 (15.8) | 53.6 (15.3) | 56.1 (16.4) | 57.5 (12.4) | 56.9 (11.8) | 58.5 (13.2) |
| Male (%) | 504,527 (47.4) | 262,458 (43.7) | 242,069 (91.1) | 372,168 (92.6) | 182,467 (49.0) | 189,701 (51.0) | 40,682 (90.1) | 23,971 (58.9) | 16,711 (41.1) | 91,677 (86.1) | 56,020 (61.1) | 35,657 (38.9) |
| Female (%) | 49,074 (5.8) | 32,265 (3.0) | 16,809 (8.9) | 29,837 (7.4) | 18,976 (63.6) | 10,861 (36.4) | 4,449 (9.9) | 2,943 (66.2) | 1,506 (33.9) | 14,788 (13.9) | 10,346 (70.0) | 4,442 (30.0) |
| BMI (SD) | 29.8 (5.8) | 30.3 (6.0) | 29.1 (5.6) | 29.7 (6.2) | 30.3 (6.4) | 29.1 (5.9) | 30.6 (6.0) | 31.0 (6.0) | 30.1 (6.1) | 30.1 (6.6) | 30.5 (6.5) | 29.4 (6.6) |

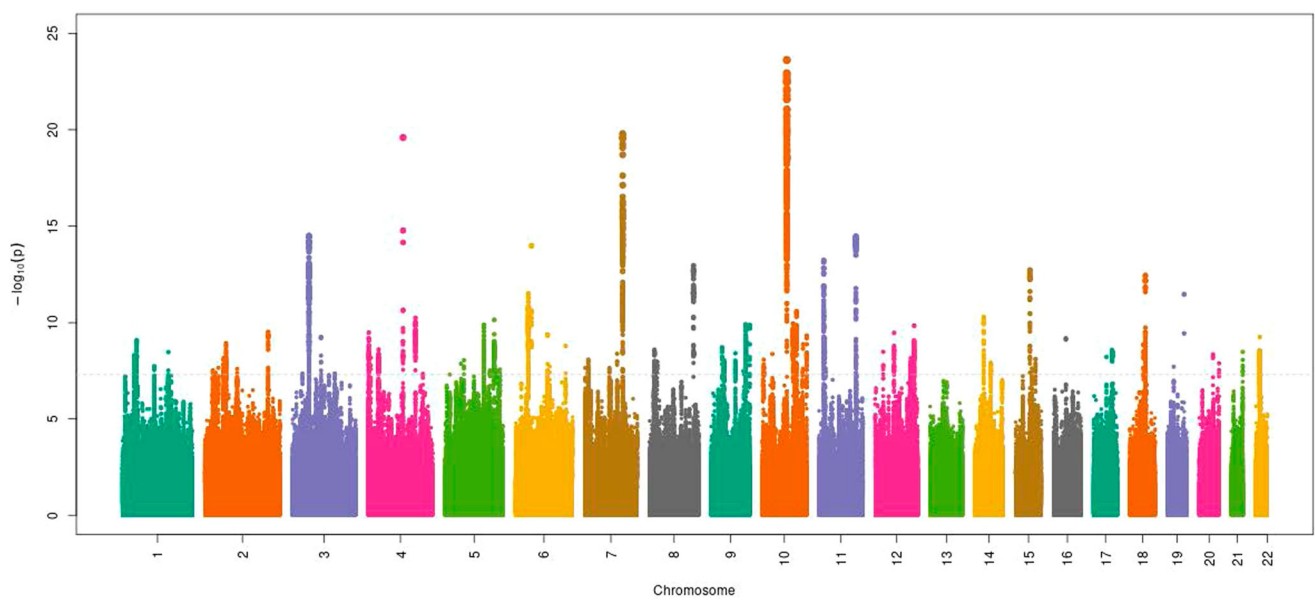

**Fig. 1 | Multi-ancestry Manhattan plot of meta-analysis GWAS of EHR defined CBP-MVP.** The $y$ axis represents -$\log_{10}$ $p$-values from two-sided $z$-test for meta-analyses effect estimates. The dotted line represents genome-wide significance at $p < 5 \times 10^{-8}$.

**Table 2 | Prior published back pain phenotypes GWAS studies**

| Publication # | Author | # Loci | Back Pain Phenotype | Year | PMID | Ref. |
|---|---|---|---|---|---|---|
| 1 | Bjornsdottir et al. | 27 | IDD | 2022 | 35110524 | 17 |
| 1 | Bjornsdottir et al. | 14 | Spinal pain ("dorsalgia") | 2022 | 35110524 | 17 |
| 2 | Belonogova et al. | 2 | CBP | 2022 | 36448979 | 73 |
| 3 | Bortsov et al. | 13 | CBP | 2022 | 35975136 | 74 |
| 4 | Suri et al. | 1 | LSRS | 2021 | 33729212 | 10 |
| 4 | Suri et al. | 1 | LSS | 2021 | 33729212 | 10 |
| 5 | Freidin et al. | 2 | Male CBP | 2021 | 33021770 | 75 |
| 5 | Freidin et al. | 7 | Female CBP | 2021 | 33021770 | 75 |
| 6 | Tsepilov et al. | 22 | Back Pain | 2020 | 32587327 | 18 |
| 7 | Freidin et al. | 5 | Back Pain | 2019 | 30747904 | 8 |
| 8 | Freidin et al. | 1 | LDD | 2019 | 30808802 | 76 |
| 9 | Suri et al. | 3 | CBP | 2018 | 30261039 | 9 |
| 10 | Williams et al. | 2 | LDD | 2012 | 22993228 | 77 |

*IDD* Intervertebral disc degeneration, *CBP* chronic back pain, *LSRS* lumbosacral radicular syndrome, *LSS* Lumbosacral Syndrome, *LDD* Lumbar Disc Degeneration.

variants according to FUMA in the multi-ancestry meta-analysis. Sixty-seven of these are novel loci that have not been previously reported as genome-wide significant and 20 were reported in prior GWAS (Supplementary Data 13).

### Novel loci and genes
The five most significant novel loci discovered in the current multi-ancestry meta-GWAS are summarized below and the remaining loci can be viewed in Supplementary Data 13.

The most significant novel variant SNP is rs12533005 (chr7:114416000, $p = 1.61 \times 10^{-20}$, OR = 0.96 (95% CI: 0.95–0.97) EA = C, EAF = 0.39). This variant is in an intron of the *FOXP2* gene and is an eQTL for *FOXP2* in three tissues in GTExs: Adipose-Subcutaneous, Pituitary, and Adrenal Gland. The second most significant novel variant is rs13107325 (chr4:102267552, $p = 2.55 \times 10^{-20}$, OR = 1.07 (95% CI: 1.06–1.09), EA = T, EAF = 0.076). It is a missense SNP in the *SLC39A8* gene and also an eQTL for the *UBE2D3* gene in Artery-Aorta tissue in GTEx. The third most significant novel variant SNP is rs2298526 (chr11:112981742, $p = 3.42 \times 10^{-15}$, OR = 0.97 (95% CI:0.96–0.98), EA = T,

EAF = 0.59) in an intron of the *NCAM1* gene and an eQTL for *NCAM1* in four tissues in GTExs: Muscle-Skeletal, Skin - Not Sun Exposed (Suprapubic), Artery - Tibial, and Skin - Sun Exposed (Lower leg). The fourth most significant novel variant SNP is rs11596214 (chr10:104694074, $p = 2.69 \times 10^{-11}$, OR = 0.97 (95% CI: 0.96–0.98), EA = A, EAF = 0.37). It is an intron of the *SORCS3* gene and a GTEx eQTL for *SORCS3* in the Pituitary and Nerve Tibial tissues. The fifth most significant novel variant SNP is rs36030569 (chr14:46832654, $p = 5.29 \times 10^{-11}$, OR = 1.03 (95% CI: 1.02–1.03), EA = A, EAF = 0.53). It is intergenic between the genes *RPL10L* and *MDGA2*. rs36030569 is a GTEx eQTL for *MDGA2* in three Brain tissues: Cortex, Nucleus accumbens (basal ganglia), and Caudate (basal ganglia). rs12533005, rs11596214 and rs36030569 replicate at Bonferroni significance in Bjornsdottir et al. rs13107325 replicates at nominal significance and rs2298526 was not available in Bjornsdottir et al.[17].

### Heritability
As described in the Methods section, all liability scale heritability estimates assumed the same sample population prevalence (0.487).

Supplementary Data 5 provides a complete summary of heritability estimates. The observed scale heritability among MVP participants of EUR ancestry was 0.059 (SE 0.0021) and the liability scale heritability was 0.093 (SE 0.0033), given the 0.501 sample prevalence. Among MVP participants of AFR ancestry, observed scale heritability was 0.045 (SE 0.0058) and liability scale was 0.075 (SE 0.0098), with a sample prevalence of 0.623. Among MVP participants of HIS ethnicity, observed scale heritability was 0.040 (SE 0.0071) and liability scale heritability was 0.066 (SE 0.012), with a sample prevalence of 0.596. In the GWAS of spinal pain in Europeans by Bjornsdottir et al.[17], observed scale heritability was 0.015 (SE 0.0006), and liability scale heritability was 0.058 (SE 0.0024), with a sample prevalence of 0.11. There was no significant difference between the observed heritability for AFR and HIS strata ($p = 0.62$, two tailed test). EUR participants had significantly greater heritability than either AFR ($p = 0.02$) or HIS participants ($p = 0.01$).

## Genetic architecture

Genetic correlations ($r_g$) were estimated between the three ancestry strata from the GWAS of CBP-MVP. For the EUR stratum compared to the AFR and HIS strata, $r_g$ for CBP-MVP were 0.56 (SE 0.064, $p = 9.7 \times 10^{-19}$) and 0.85 (SE 0.095, $p = 3.4 \times 10^{-19}$), respectively. Genetic correlations between the African and Hispanic strata were $r_g = 0.87$ (SE 0.17, $p = 1.6 \times 10^{-7}$). The EUR stratum of the GWAS of CBP-MVP had a

high genetic correlation $r_g = 0.91$ (SE 0.019, $p < 10^{-274}$) with the GWAS of spinal pain by Bjornsdottir et al., which included Europeans only[17]. Genetic correlations between the AFR and HIS strata of the current GWAS of CBP-MVP and the GWAS of spinal pain by Bjornsdottir et al. were 0.54 (SE 0.066, $p = 5.7 \times 10^{-16}$) and 0.74 (SE 0.093, $p = 2.1 \times 10^{-15}$), respectively.

Next, we extended our analyses with MiXeR, which uses univariate and bivariate Gaussian mixture modeling to estimate polygenicity and discoverability. MiXeR estimated ~11,700–11,800 independent risk loci in MVP EUR participants, with ~42,100–60,200 in HIS and ~148,100–248,500 in AFR. HIS, and EUR genetic overlap was ($r_g = 0.44$; ~6000 shared risk loci overlap) where AFR in comparison to EUR ($r_g = 0.30$; ~10,400 loci overlap) and HIS ($r_g = 0.34$; ~50,000 loci overlap) (See Supplementary Fig. 8) also showed considerable overlap. *Post-hoc* power to detect all risk loci was much higher at lower sample counts in EUR (80% power at ~31,700,000 participants) than either HIS (80% power at ~50,000,000 participants) or AFR (73% power at ~79,000,000 participants) (See Fig. 2).

The power to detect all risk loci was much higher at lower sample counts in EUR (80% power at ~31,700,000 participants) than either HIS (80% power at ~50,000,000 participants) or AFR (73% power at ~79,000,000 participants).

We additionally examined the genetic overlap between MVP EUR ancestry men and women with MiXeR using bivariate analysis (Fig. 3). All the predicted men's variation overlap with women's variation. The EUR men-specific estimated variation causal of backpain nearly matches the full cohort EUR estimate. The women-specific estimated causal variation is nearly six times that of the men, but the smaller sample size of the women in the cohort has a large standard error estimate that overlaps the men's estimate and makes drawing conclusions from the women-specific variation uncertain.

In general, genetic correlations between the current GWAS of CBP-MVP in participants of EUR ancestry and previously reported GWAS of self-reported pain phenotypes[18] were of large magnitude ($r_g = 0.85$, $p < 10^{-274}$ for back pain; $r_g = 0.73$, $p = 2.2 \times 10^{-205}$ for neck pain; $r_g = 0.69$, $p = 1.6 \times 10^{-153}$ for hip; $r_g = 0.53$, $p = 2.5 \times 10^{-97}$ for knee pain). See Supplementary Data 14, 15 for a complete summary of LDSR $r_g$ estimates of ancestry strata and other backpain comparisons.

When examining 823 genetic correlations between the GWAS of CBP-MVP in participants of EUR ancestry, we found 491 significant correlations ($p < 6.08 \times 10^{-5}$) with reported traits. Correlations were strongest within the pain, neurological, socioeconomic, and musculoskeletal domains (Fig. 4, all genetic correlation statistics are also summarized in Supplementary Data 16). The top individual results

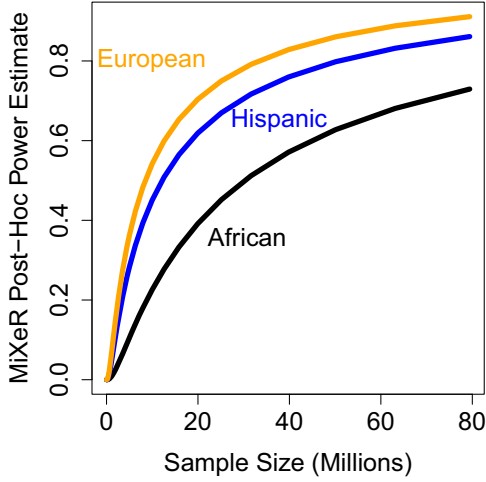

**Fig. 2 |** Post-Hoc MiXeR power estimates in MVP EUR, HIS and AFR strata.

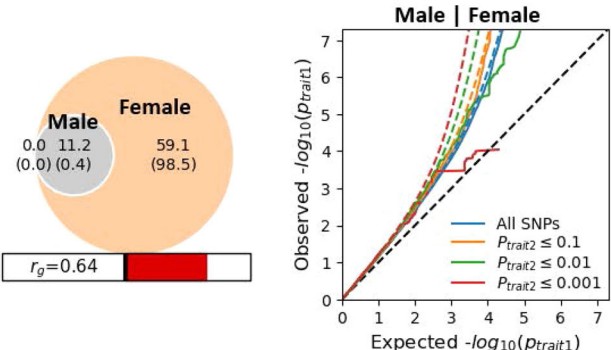

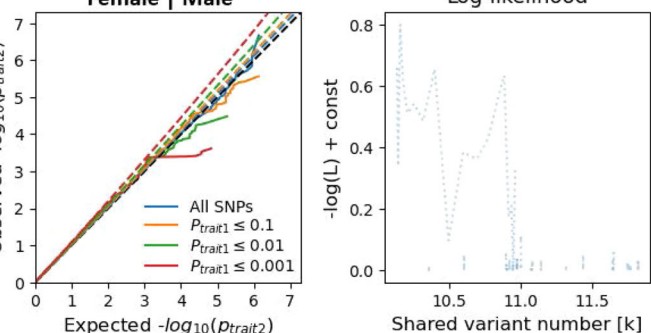

**Fig. 3 | Genetic overlap between EUR ancestry male and female participants, using MiXeR.** Venn diagram reports numbers in thousands of variant risk loci (standard error) related to the first trait (gray circle), second trait (orange), and the mutually shared variants (gray overlap). Genetic correlation is depicted as a number and red progress bar. The two center plots show conditional QQ plots of

trait 1 on subset 2 and trait 2 on subset 1. The model log-likelihoods based on the number of causal variants are shown in the rightmost plot. The solid blue line reflects the average likelihood across the 20 MiXeR runs. The dotted blue line reflect the likelihood of individual runs.

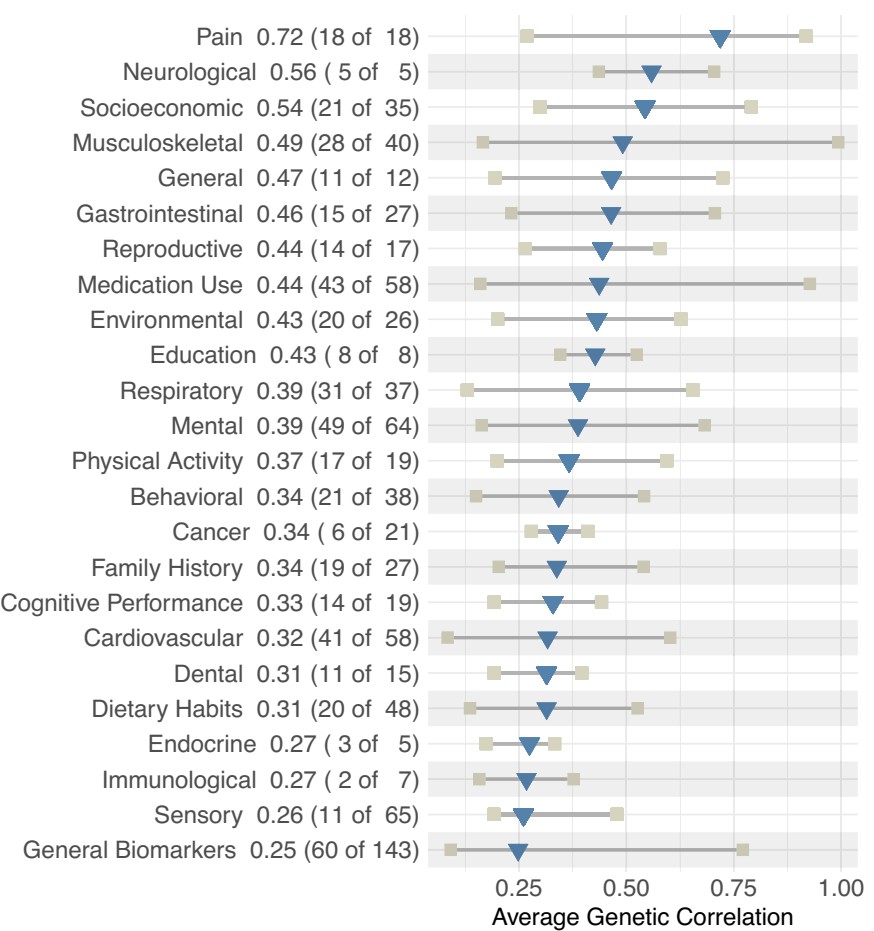

**Fig. 4 | Summary of 491 significant LDSC genetic correlations ($r_g$s) between EUR MVP-CBP and various traits presented by domain.** $R_g$ is presented next to domain name, number of significant correlations out of total number per domain presented parenthetically. Triangles represent the average $r_g$ per domain, and squares represent the minimum and maximum $r_g$ within the domain.

were related to back pain phenotypes, sub-phenotypes, or treatments, including participant-reported back pain ($r_g = 0.89$, SE = 0.02, $p < 10^{-274}$), participant-reported multisite chronic pain ($r_g = 0.84$, SE = 0.02, $p < 10^{-274}$), EHR-defined dorsalgia (see Bjornsdottir et al.[17] above) ($r_g = 0.92$, SE = 0.05, $p = 3.2 \times 10^{-79}$), and EHR-defined spondylopathies ($r_g = 0.99$, SE = 0.09, $p = 1.8 \times 10^{-26}$). Correlations were also large for proxies for disability like "long-standing illness, disability, or infirmity" ($r_g = 0.65$, SE = 0.03, $p = 4.3 \times 10^{-138}$) and conventional back pain risk factors like BMI ($r_g = 0.35$, SE = 0.02, $p = 5.8 \times 10^{-80}$). Our findings also reveal notable correlations with psychiatric disorders, specifically depression ($r_g = 0.62$, SE = 0.02, $p = 1.3 \times 10^{-211}$), major depressive disorder ($r_g = 0.50$, SE = 0.02, $p = 1.3 \times 10^{-112}$), attention deficit hyperactivity disorder ($r_g = 0.60$, SE = 0.03, $p = 4.3 \times 10^{-85}$), and treatment with amitriptyline ($r_g = 0.89$, SE = 0.09, $p = 1.5 \times 10^{-21}$).

### Expression and functional characterization

FUMA Magma GTEx tissue analysis of the loci significant in the multi-ancestry analysis showed highly significant ($p - 10^{-7}$) enrichment of eQTLs in Brain and Pituitary tissues (Supplementary Fig. 9). Nominally significant ($p < 0.05$) enrichment of eQTLs was observed in cervix uteri, nerve, and uterine tissues (See Supplementary Data 18, 19). Among the 90 significant lead variants from the multi-ancestry meta-GWAS, 65 are reported as eQTLs for one or more genes in GTEx v8 (See GTEx annotation in Supplementary Data 17). Multi-ancestry FUMA analysis identified 816 eQTLs among all the significantly CBP-MVP lead associated genetic variants (See Supplementary Data 23).

Figure S10 presents results from cell type analyses conducted using the FUMA Cell Type module and the "GSE67835_Human_Cortex_woFetal" dataset through MAGMA cell specificity analyses. Panel A highlights cell type specificity, showing neurons, astrocytes, oligodendrocytes, and microglia with neurons having the highest specificity. Panel B identifies significant cell types across datasets, reinforcing the relationship with neurons and glial cells (e.g., astrocytes). Panel C shows within-dataset conditional analyses, isolating the independent contributions of each cell type with significant associations for neurons and glial cells. Panel D shows pair-wise cross-datasets conditional analyses, showing conditional dependencies and collinearities between cell types, with neurons and glial cells again showing significant associations. These results underscore the importance of specific neuronal and glial cell types in the genetic architecture of CBP.

### Mendelian randomization of discovery eQTLs

We examined the possible causal roles of the 62 lead variants from the discovery GWAS of CBP-MVP which were eQTLs reported in GTEx. Using single variant two-sample MR, 48 variants had significant (FDR < 0.1) causal associations with spinal pain "Dorsalgia" reported in Bjornsdottir et al.[17]. Among the eQTLs with significant causal associations with spinal pain, four genes (*INPP5B, DRD2, HTT, SLC3OA6*) had targets with known drug compounds annotated in Drugbank[19] that have activity on the gene product (See Table 3 for a summary of the genes' annotated drug compounds and Supplementary Data 20 for the MR estimates of 62 variants with

**Table 3 | MR significant (FDR < 0.1) eQTL genes with drug compounds annotated in Drugbank[19] known to have activity on the gene product**

| Gene | Drug Compounds |
|------|----------------|
| INPP5B | D-Myo-Inositol-1,4-Bisphosphate |
| DRD2 | Cabergoline, Ropinirole, Sulpiride, Promazine, Prochlorperazine, Droperidol, Chlorpromazine, Haloperidol, Fluphenazine, Apomorphine |
| HTT | Copper |
| SLC30A6 | Zinc chloride, Zinc sulfate |

Bjornsdottir et al outcome summary statistics of the 65 MVP-CBP lead variants that are eQTLs in GTEx).

## Discussion

In this multi-ancestry meta-GWAS conducted in the diverse MVP cohort, we identified 87 loci associated with CBP, of which 67 loci were novel. Among the 90 independent lead variants at these loci, 57 were replicated in an independent GWAS of European participants using a related spinal pain phenotype[17] at the nominal threshold of significance (<0.05), while 26 replicated at the Bonferroni-corrected threshold of significance. These findings more than double the number of previously identified loci ($n = 59$) with variants ($n = 94$) associated with CBP at the level of genome-wide significance. A few novel variants have been identified in related pain phenotypes, such as chronic multisite pain and rheumatoid arthritis, but they have not been reported specifically in CBP[18,20].

A notable finding of the current meta-GWAS of CBP-MVP ($n = 553,601$) is the large number of independent variants and loci found exceeding the threshold for genome-wide significance. The largest GWAS of a spinal pain phenotype to date by Bjornsdottir et al. included nearly twice the sample size ($n = 1,028,947$ individuals) identified 41 independent variants at 33 loci. The current meta-GWAS of CBP-MVP has a balanced case-control proportion (~53%) and a larger number of cases than in the study by Bjornsdottir et al.[17]. The MVP cohort may be a more favorable setting for back pain-related genetic discovery. Military service is associated with many of the physical and occupational environmental exposures that have historically been associated with back pain, such as frequent lifting and carrying, prolonged periods in uncomfortable positions, and others[21,22]. However, military service is also associated with unique exposures specific to combat, such as wearing body armor for prolonged periods and major physical trauma. Additionally, the psychosocial risk factors which are widely accepted as the strongest risk factors for CBP, such as depression and other mental health comorbidities[21–23], are highly prevalent in military Veterans[7]. This setting of a high prevalence of environmental risk factors for CBP may have unmasked genetic predispositions and facilitated genetic discovery in the MVP cohort.

Of the 67 novel lead variants identified in the current meta-GWAS of CBP-MVP, the five most significant were eQTLs. Two of these variants are eQTLs in nerve (rs11596214, SORCS3) and brain tissues (rs36030569, MDGA2), suggesting possible roles involving the nervous system and pathways related to pain sensation and/or perception. rs2298526 is an eQTL for NCAM1 in skeletal muscle, an important component of the musculoskeletal system that is often implicated in CBP[24]. NCAM1 regulates neuron-neuron and neuron-muscle interactions through neurite outgrowth[25]. rs13107325 has been shown to be associated with pain related phenotypes of knee osteoarthritis[26], rotator cuff disease[27] and adolescent idiopathic scoliosis[28–30]. The possible connections of other lead variants to CBP are less evident. FOXP2 (rs12533005, eQTL) has known causality to language disorders[31]. rs13107325 is a missense SNP in the SLC39A8 gene and also an eQTL for the UBE2D3 in Artery-Aorta tissue. SLC39A8 is a zinc and manganese transporter. Follow-up of the 62 eQTLs that were significant in the multi-ancestry discovery GWAS, using two-sample MR conducted in independent samples, identified potentially causal associations for 48 of these eQTLs.

Given the larger number of variants included in contemporary GWAS as compared to when the conventional $p$-value threshold for GWAS ($p < 5 \times 10^{-8}$) was first adopted as a standard[32] it has been suggested that $3 \times 10^{-8}$ be adopted as the new threshold for studies that investigate variants with MAF as low as 1%[33]. Adopting this more conservative threshold would have resulted in 2977 being declared genome-wide significant in the multi-ancestry meta-analysis, instead of 3444 variants using the $p$-value threshold of $p < 5 \times 10^{-8}$. Similarly, of the lead variants reported among these, 80 of the 90 total we report would remain genome-wide significant.

We identified four MR associated eQTLs with corresponding drug compounds annotated in Drugbank. Based on these GWAS results, these findings can potentially identify drugs for repositioning to provide novel lead compound therapies in clinical trials. Notably, our analysis identified dopamine receptor D2 (DRD2)-targeting drugs, such as Apomorphine, Cabergoline, and Haloperidol, which falls in line with a limited number of trials supporting the idea that dopaminergic interventions—predominantly used for conditions such as Parkinson's disease, schizophrenia, and bipolar mood disorder—can also change various aspects of pain perception. Dopamine receptors, including DRD2, have been evaluated in pain modulation, and DRD2 agonists and antagonists have been shown to influence pain thresholds[34,35]. For example, Apomorphine, a short-acting dopamine agonist, has been shown to increase the ability to tolerate cold pain in patients with chronic lumbar radiculopathy[34]. Furthermore, imaging results demonstrate that individuals with CBP showed differences in dopamine function that aligned with pain sensitivity, effect, and endogenous opioid system activity[35]. Given the multiple DRD2 drugs identified by our results ($n = 10$) and the known implications with reward-based behaviors that dopamine controls along with mental health[36], this group of drugs may be the most useful to follow up. Support for other targets is less clear but not without potential. Studies of inositol phosphates have demonstrated links to neuronal signaling, metabolic disorders, and inflammatory function[37]. Zinc has been linked to pain modulation[38], and animal studies have demonstrated that zinc deficiency can exacerbate pain while supplementation may have analgesic effects[39]. Copper is involved in dopamine to norepinephrine conversion[40] and copper metabolism disorders, typically copper excess, have been associated with neurological symptoms that may include pain[41,42]. Furthermore, the balance between copper and zinc is important as they can antagonize each other's absorption and function, with imbalance potentially disrupting neurological function and exacerbating pain. For example, a higher copper-to-zinc ratio has been found in patients with rheumatoid arthritis[43] and low zinc in individuals with fibromyalgia[44]. While little is known about group differences in these associations, future research can evaluate these relationships in larger genetically informative samples.

To our knowledge, this is the first multi-ancestry GWAS of back pain to include large numbers of participants from different ancestry groups (>30,000 individuals within ancestry strata). The diversity of the MVP cohort reveals differences in CBP prevalence and heritability between ancestry and ethnicity groups. While the prevalence of CBP-MVP in the current study was 53.2%, comparable to the CBP population prevalence assumed a priori (48.7%) and the prevalence of CBP in a prior GWAS of 100,811 adults from 10 United States health systems (48.8%), we found a substantially higher prevalence of CBP in MVP participants of African ancestry (62.3%) and Hispanic ethnicity (59.6%), as compared to participants of European ancestry (49.9%). These differences in CBP prevalence by ancestry and ethnicity contrast with prior cross-sectional population-based surveys which suggest no differences in general chronic pain prevalence (not limited to CBP specifically) between race and ethnicity groups[45,46], but are consistent with

studies in clinical populations, which have shown higher rates of continuing CBP among patients who self-report as Black or African-American or Hispanic[47–49]. In the same studies in clinical populations, accounting for environmental factors such as proxies of socioeconomic status reduced or eliminated differences in CBP according to race and ethnicity[46–48].

Alongside differences in CBP-MVP prevalence by ancestry in the current study, genetic contributions to CBP-MVP as measured by SNP-based heritability were higher in participants of EUR ancestry (liability scale heritability 9.3%) as compared to AFR ancestry (7.5%) and HIS ethnicity (6.6%). As heritability estimates can decrease in settings where there is a greater influence of environmental factors (including social factors)[50], the current findings suggest social and other environmental differences as likely factors explaining differences in prevalence and heritability of CBP-MVP by ancestry groups. It is worthwhile to note that the current analysis in the MVP did not account for such environmental factors due to the study's focus on genetic predictors of CBP, and the potential for covariate adjustment for environmental factors in GWAS to introduce statistical bias[51]. However, future epidemiologic studies of CBP prevalence in Veterans by ancestry, race and ethnicity groups should examine these relationships after accounting for proxies of socioeconomic status, as well as other important factors such as service history and comorbidities.

We did not detect GWS variants in the Hispanic strata. This is likely due to limitations in power given the smaller sample size of this stratum. Thirty-three of the GWS meta-analysis hits had nominally significant results in the Hispanic strata. The strata specific results can be browsed in the Supplementary Data 21–25. There were significant genetic correlations between CBP-MVP in the HIS stratum and the EUR ($r_g = 0.85$) and AFR strata ($r_g = 0.87$). Newer GWAS methods such as TRACTOR[52] would enable the HIS strata to be deconvoluted to the contributing Native American and primarily introgressed European, African, Asian, and Pacific Islander populations. Similarly, many African Americans are admixed and would benefit from ancestry deconvolution. Remarkably, the MVP AFR stratum has a lower genetic correlation with the EUR stratum (MVP $r_g = 0.56$ and Bjornsdottir et al. $r_g = 0.54$), suggesting different genetic contributions to back pain among African and European American Veterans. Genetic overlap, estimation of the number of causal risk loci and *post-hoc* power using MiXeR allowed for more detailed comparisons of ancestry. Much tighter estimation of the total number of risk loci was evident for the EUR compared to the AFR and HIS strata in the estimations of bivariate overlap between ancestries. This is likely due to the majority sample size of the MVP being of EUR ancestry reflecting the prevalent demographics of the United States. The total number of risk loci was lower and the power to detect at lower sample size was higher overall in EUR than HIS or AFR. Many more risk loci are estimated in AFR and HIS, even with their significantly lower observed heritability LDSC regression. This suggests that HIS and particularly AFR have many more causal loci that likely have lower effect sizes and are less penetrant and/or are lower frequency than EUR risk loci. Roughly half of the EUR risk loci are shared between either HIS or AFR risk loci. More detailed local genetic overlap analyses could be performed with a tool like SUPERGNOVA[53], but the underpowered sample sizes in the HIS and AFR strata would likely give misleading results and underestimate local genetic overlaps, as only one of two genome-wide significant variants detected replicates in these strata, in an apparently somewhat AFR specific locus.

Our findings reveal strong genetic correlations between EHR-defined CBP in MVP and various other phenotypes, including self-reported and/or EHR-defined back pain in other studies and multisite chronic pain. There is a large overlap with the 823 phenotypes interrogated for genetic correlations with CBP-MVP, of which 491 met the Bonferroni significance threshold. Strong positive genetic correlations were observed with several psychiatric and mental health related phenotypes. Treatment with the tricyclic antidepressant amitriptyline has a genetic correlation ($r_g = 0.89$) comparable to back pain phenotypes (Bjornsdottir et al.[17], $r_g = 0.92$), which may reflect shared genetic influences with depression and/or that amitriptyline is sometimes used as a CBP treatment. Chronic pain can be both a cause and consequence of mental health disorders, and our findings underscore the contributions of shared genetic components to the chronic pain-mental health relationship.

This study has several limitations. While this is the first GWAS of back pain to include large numbers of participants from multiple ancestry and ethnicity groups, some such groups within MVP had small sample sizes ($n < 5000$) and for this reason were not analyzed. This is an important limitation that can be addressed in the next few years as the more diverse 'All of Us' genomic biobank increases in size. We analyzed the 65 most significant eQTLs for their causal relationship with spinal pain[17] and found that 48 of them had significant causal associations using MR. This does not preclude that other collinear eQTLs regulated in the same epigenetic machinery as seen among the total 816 gene eQTLs identified by FUMA are not the true causal genes affected by these 65 variants. We chose to only analyze the most significant gene IVs from these loci to prioritize the importance of the loci over the specific genes that are co-regulated in a unit with the same variant eQTL. This maximizes power at determining the causality of the loci at the expense of knowing which gene product may be truly causal. This also means that the compounds we annotate from Drugbank[19] may not have efficacy due to being the wrong target protein. Another limitation is the size and case ratio of the replication GWAS by Bjornsdottir et al.[17]. Despite being larger in total sample size, it has fewer cases and identifies fewer significant variants and loci than the discovery GWAS of CBP-MVP. As a result, we were less likely to replicate as many of our CBP-MVP significant lead variants ($n = 26$ replicated) when using a Bonferroni correction. To mitigate this, we also reported nominally ($p < 0.05$) significant variants ($n = 57$). A third limitation of the replication cohort is that it did not use exactly the same phenotype as CBP-MVP but one that is closely related, as evidenced by high genetic correlations (up to $r_g = 0.91$) between the two phenotypes. The 48 variants that show significant causal associations with spinal pain after accounting for FDR in the MR analysis of the lead variant eQTLs provide further support for the associations with CBP-MVP. A final limitation of the current study is that while it cross-referenced CBP-MVP-associated loci with prior GWAS of back pain-related phenotypes, it did not do so for GWAS of other distinct pain phenotypes such as symptomatic knee osteoarthritis[21], pain intensity irrespective of pain site[54], etc. Our genetic correlation results indicate substantial genetic overlap with numerous other pain phenotypes. Future work should more deeply study the overlapping and distinct loci between CBP-specific pain loci and those of loci implicated in generalized pain perception and other pain phenotypes.

The novel and previously known genetic risk information identified in the current GWAS of CBP may target proteins for pain drug development, inform personalized medicine approaches, and provide a path to improve treatment options for those suffering from CBP with a better understanding of the genetic etiology of CBP. Here we show this through MR analyses that prioritized among the many lead variants and identified four loci with known compounds targeting them. The most biologically interesting among these is a dopamine receptor (DRD2) with known reward and mental health etiology. This work substantially contributes to the body of knowledge surrounding back pain genomics by the sheer number of novel loci identified. Future work on the genomics of CBP should be focused on predictive models that integrate genomics with clinical and occupational histories, spinal imaging, lead drug target identification, and biomarkers to increase the utility of precision medicine[55].

## Methods

### Study participants and design

The current observational study was conducted in the Million Veteran Program (MVP), a national research project to determine how genetic traits, health habits, and environmental factors affect Veteran health and illness. The design and recruitment procedures of MVP have been documented in detail previously[15]. In brief, MVP data include self-reported survey, electronic health record (EHR), and genetic data. The MVP protocol was approved by the VA Central Institutional Review Board (cIRB) in 2010, and study enrollment began in 2011. The VA cIRB and the Research and Development Committee at VA San Diego Healthcare System approved the current analyses. With the 2022 data release, 819,417 participants were enrolled in the MVP and 662,681 had genetic data available for analyses reported here.

This study involved genetic discovery in a multi-ancestry GWAS meta-analysis of CBP conducted in the MVP cohort; replication in an independent GWAS sample; functional characterization of findings from the discovery GWAS; and examination of genetic correlations with other phenotypes.

### GWAS cohort and phenotype

Chronic back pain in MVP (CBP-MVP) was assessed as a binary variable, and case and control definition information were derived using EHR ICD 9/10 codes. ICD codes 724.2, M54.5, M54.89, M54.42, M54.41, and M54.40 were used to identify potential cases of CBP (Supplementary Data 1). Coding algorithms for CBP using ICD codes have been previously validated and widely used in back pain research over the past three decades[10,56–59]. CBP-MVP cases were defined as Veterans with two or more ICD codes appearing at least 90 days apart. Ninety days, approximately 3 months, is the minimum duration of time with back pain that is typically used to separate CBP (≥3 months) from acute back pain (<3 months)[60]. Controls were defined as Veterans who did not have at least two ICD codes for back pain. Out of 819,417 enrolled Veterans, individuals were excluded if they did not have a coded gender ($n = 121$), had only one occurrence of back pain ($n = 86,864$), or had back pain diagnoses less than 90 days apart ($n = 23,904$), were not genotyped ($n = 139,038$) or small or missing ancestry group ($n = 15,889$). Henceforth, in this manuscript, the abbreviation "CBP-MVP" refers to the phenotype examined in the MVP cohort, and "CBP" refers to chronic back pain more generally.

### Genotyping, imputation, and quality control

Genotyping, imputation, and quality control within MVP have been described previously and were conducted by the MVP project core working group. Briefly, MVP samples were genotyped using a 723,305 single nucleotide polymorphism (SNP) Affymetrix Axiom Biobank array, customized for MVP to include variants of interest in multiple diverse ancestries[15]. Imputation was performed with Minimac4 using TopMed reference panel data. The analyses were performed using MVP Release 4 data (GRCh38). Final genotype data consisted of 96 million genetic variants. Principal components were calculated for each ancestry using PLINK 2.0 alpha[61] on genotyped data.

### Race, ethnicity, and ancestry

Harmonizing Genetic Ancestry and Self-identified Race/Ethnicity (HARE) groups were used to define race/ethnicity[62]. Briefly, HARE enhances classification by integrating self-identified race/ethnicity (SIRE) and genetically inferred ancestry (GIA). HARE ensures accurate classification by using GIA to refine and, if necessary, impute SIRE, improving the reliability of race/ethnicity assignment in genetic research. Less than 2% of individuals are not assigned a HARE group when participant-identified and genetically-inferred ancestry data

produce discordant results. Here, we use the term "Hispanic" for the HARE race and ethnicity groups comprised of individuals who are Latino or Hispanic, the term "European" for individuals who are White but not Hispanic, and "African" for individuals who are Black but not Hispanic.

A total of 553,601 participants had available phenotype and genotype information and were used for GWAS analysis (Table 1). The HARE sample sizes for European ancestry (EUR, $n = 402,005$ total; 201,443 cases, 200,562 controls), African ancestry (AFR, $n = 106,465$ total; 66,366 cases, 40,099 controls), and Hispanic ethnicity (HIS, $n = 45,131$ total; 26,914 cases, 18,217 controls) groups were used in this analysis. People of East Asian and South Asian ancestry were not analyzed due to the low numbers of Asian individuals in MVP. Hereafter, EUR, AFR, and HIS refer to the HARE-defined ancestry strata in the MVP.

### Computation and statistical analysis

**GWAS regression.** GWAS analysis was conducted for the three ancestry groups separately using logistic regression to test the association between CBP and imputed dosages using REGENIE v2.2.26[63]. REGENIE is a two-step machine learning method that adjusts for relatedness. Step One analyzed MVP genotype array data only, dividing SNPs into blocks, and used ridge regression to generate predictions. These predictions were combined in a second ridge regression and decomposed by chromosome for leave-one-chromosome-out analysis, which served as covariates in Step Two. Step Two used Release 4 imputed data for cross-validation and applied Firth logistic regression and saddle point approximation for the binary trait analysis. Analyses were conducted for all participants of EUR, AFR, and HIS HARE ancestries, with the models including the first 10 principal components of genotype and self-reported gender as covariates. Additionally, separate analyses were carried out for EUR men and EUR women, using models that included the first 10 principal components as covariates. SNPs with an imputation INFO score > 0.6, minor allele frequency (MAF) ≥ 0.01, and HWE > $1 \times 10^{-3}$ were reported in the analysis. For the primary analysis, a genome-wide significance threshold was set as a $p \leq 5 \times 10^{-8}$.

**Meta-analysis.** Meta-analysis was conducted across three ancestries (EUR, AFR, HIS; $n = 553,601$) using the METAL software package[64] with default parameters. The results of each study were weighted based on the square root of its sample size. After filtering for MAF ≥ 0.01 and correcting inconsistent allele labels and strands, 17,466,242 variants remained. Cochran's Q-test was performed for each SNP to test for heterogeneity of effect. We applied an FDR correction across all variants meta-analyzed and then inspected (FDR < 0.1) lead variants at each genomic locus for heterogeneity of effect.

**FUMA.** Functional Mapping and Annotation of Genome-Wide Association Studies (FUMA) was used[65–67] to annotate variant results. The MAGMA v1.06 method[68] and the MsigDB v5.2 database[69] from FUMA were used. FUMA results are reported in GhGRC37 (hg19). The default settings were used in all FUMA analyses. The SNP2Gene module defined independent genomic risk loci and variants in LD. Independent significant SNPs dependent at ($r^2 \geq 0.1$) assigned to the same locus, and independent significant SNPs closer than 250 kb were merged into same locus. Each locus was represented by the top lead SNP with the lowest p-value in the locus. Ancestry-appropriate reference panels were used (EUR for EUR, AFR for AFR, and AMR for HIS). SNPs were mapped to protein-coding genes. Functional consequences of SNPs were derived from mapping SNPs on their chromosomal position and reference alleles to reference databases containing annotations, including ANNOVAR, Combined Annotation Dependent Depletion (CADD), RegulomeDB (RDB), and chromatin

states in tissues/cell types. FUMA Cell Type module with the GSE67835_Human_Cortex_woFetal dataset was used to identify brain cell types associated with CBP.

**Replication.** To replicate genome-wide significant SNPs, we used publicly available meta-GWAS summary statistics from an independent, non-overlapping contemporary meta-GWAS of the EHR-defined spinal pain "dorsalgia" phenotype by Bjornsdottir et al.[17] ($n = 119,100$ cases, $n = 909,847$ controls). This phenotype is similar to the CBP-MVP phenotype but includes cases of neck and back pain; we refer to this subsequently as the "GWAS of spinal pain". The GWAS of spinal pain used for replication included samples from deCODE Genetics (Iceland), the Danish Blood Donor Study (DBDS), Copenhagen Hospital Biobank (CHB), the UK Biobank (United Kingdom), and FinnGen (Finland). For analyses in the replication sample, the threshold for statistical significance was determined using a Bonferroni correction, with the conventional threshold of nominal significance ($p < 0.05$) divided by the number of SNPs studied in the replication sample. We also report those associations that reach nominal ($p < 0.05$) in the replication sample. Genetic correlations (rg) between CBP-MVP and publicly available spinal pain-related summary statistics[18] (including the GWAS of spinal pain by Bjornsdottir et al.)[17] were calculated to allow comparison of these phenotypes.

**Linkage disequilibrium score regression and SNP-based heritability.** Linkage disequilibrium score regression (LDSC)[70] was used to estimate SNP-based heritability (h2) of CBP-MVP for each ancestry. We built LD score references from the current hg38 1000 Genomes references ($N = 2504$ participants) for each ancestry group using the EUR ($n = 503$, European), AFR ($n = 661$, African), and AMR ($n = 347$, Hispanic) groups and harmonized allele identities at genomic positions with the MVP GWAS while removing discordant and duplicated positions. For heritability estimation, a CBP population prevalence of 48.7% was used, consistent with prior meta-GWAS of CBP[10]. The extent to which the test statistic inflation was due to polygenic signal (rather than population stratification) was calculated with LDSC as 1 - (LDSC intercept - 1) / (mean observed $\chi^2$ - 1)[70].

**Genetic correlation with traits and between ancestry strata.** Bivariate LDSC regression was used to assess the genetic correlation of CBP-MVP between each ancestry stratum in MVP. We also assessed the genetic correlation between CBP-MVP in each of the three MVP ancestry strata and GWAS of spinal pain by Bjornsdottir et al.[17] that we used for replication. When comparisons were made between EUR and AFR or HIS strata, the EUR LD score 1000 genomes reference was used. Genetic correlations were also examined between CBP-MVP in EUR and eight other chronic pain phenotypes from independent samples, using publicly available GWAS summary statistics[17,18] as stated in the replication section above. To further elucidate the relationship between CBP-MVP and other phenotypes, we leveraged publicly available GWAS data within the Complex Trait Genetics Virtual Lab[71] (https://vl.genoma.io/). Cross-trait LDSC regression was performed on various conditions to assess their genetic correlation with CBP-MVP. To ensure robust findings, we restricted our analyses to phenotypes displaying a SNP heritability z-score greater than 4, yielding 823 phenotypes for evaluation. Bonferroni adjustment was applied to control for multiple comparisons, setting the threshold for statistical significance at $p < 6.08 \times 10^{-5}$.

**Mendelian randomization.** To further characterize the lead variants identified in the multi-ancestry discovery meta-GWAS of CBP-MVP that are known expression quantitative trait loci (eQTLs), and whether these eQTLs may have causal effects on CBP, we conducted *post hoc* Mendelian randomization (MR) analyses using the TwoSampleMR R package (v0.5.6). From among lead variants associated with CBP-MVP at the genome-wide significance threshold, we selected variants with previously reported GTEx estimates as the exposure instrument variables (IVs). We applied each of these IVs with the two-sample single variant Wald ratio method of MR analyses using the outcome GWAS summary data from the previously reported GWAS of spinal pain by Bjornsdottir et al.[17]. This analysis separates the discovery GWAS of CBP-MVP from the two-sample MR conducted in independent samples using GTEx exposure estimates and Bjornsdottir et al.[17] outcome estimates. We assessed significance at an FDR < 0.1. We then inspected these genes in DrugBank[19] and report any pharmaceutical compounds that target these lead variant eQTLs.

**Mixed effects score regression (MiXeR).** We used the mixed effects score regression (MiXeR) framework[72] to estimate the bivariate genetic overlap between three ancestry strata of MVP-CBP and between the men and women within the EUR strata. MiXeR utilizes a Bayesian approach to provide posterior probabilities for two key estimates: the number of shared and the number of trait-specific loci. This approach allows for an unbiased estimation of the genetic overlap, independent of the power of individual GWAS. MiXeR also provides estimates of polygenicity and the current power to detect causal loci using the univariate analyses of each ancestry. Analyses were conducted using MiXeR v1.3.

## Reporting summary
Further information on research design is available in the Nature Portfolio Reporting Summary linked to this article.

## Data availability
GWAS and meta-analysis summary statistics will be available in dbGaP (https://www.ncbi.nlm.nih.gov/gap/) upon publication under accession phs001672. MVP summary data access can be obtained by submitting a data access request through dbGaP; raw data are protected and are not available due to privacy reasons. Dataset from Bjornsdottir et al.[17] are available on https://www.decode.com/summarydata. FUMA provide the reference panels and datasets used in the described analysis; drug-class and drug-set analyses were done using Drugbank.com.

## Code availability
Data analysis was conducted using code from dbSNP, Regenie (https://rgcgithub.github.io/regenie/), Metal (https://github.com/statgen/METAL), LDSC, 2sampleMR (https://github.com/MRCIEU/TwoSampleMR), and MiXeR (https://github.com/precimed/mixer).

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

## Acknowledgements

Dr. Stanaway is supported by Veteran's Affairs Rehabilitation Research and Development Service Award# I01RX004291. Dr. Suri is a Staff Physician at the VA Puget Sound Health Care System and is supported by Veteran's Affairs Rehabilitation Research and Development Service Award# I01RX004291 and I01RX003248. Dr. Roseen is supported by career development award (National Center for Complementary and Integrative Health [NCCIH], K23-AT010487), which supported his work on this manuscript. Dr. Gasperi is supported by the VA Career Development Award #1IK2CX002107 from the US Department of Veterans Affairs, Clinical Science Research and Development Service. The authors gratefully acknowledge the continued cooperation and participation of the members of the Million Veteran Program; without their contribution, this research would not have been possible. This research is based on data from the Million Veteran Program (Project MVP033), Office of Research and Development, Veterans Health Administration. This publication does not represent the views of the Department of Veteran Affairs or the United States Government.

## Author contributions

Obtained funding: M.G.; clinical: P.S., M.G., and N.A.; statistical analysis: I.S., M.G., A.G., D.D., and S.P.; writing group: I.S., M.G., P.S., N.A., A.G., D.D., and E.R.

## Competing interests

The authors declare no competing interests.

## Additional information

## Million Veteran Program

**Ian B. Stanaway** [1,2]**, Pradeep Suri**[1,3,4,5]**, Daniel Dochtermann** [9]**, Armand Gerstenberger**[1,10]**, Saiju Pyarajan** [9] **& Marianna Gasperi** [1,6,7,8,10,13] ✉

A full list of members and their affiliations appears in the Supplementary Information.

