## [Peer Review File · Nature Communications]

A Multi-Ancestry Meta-analysis of Genome-Wide Association Studies Discovers 67 New Loci Associated with Chronic Back Pain: Findings from the Million Veteran ProgramReviewers' Comments:

Reviewer #1:

Remarks to the Author:

A Multi-Ancestry Meta-analysis of Genome-Wide Association Studies of Chronic Back Pain in 553,601 Veterans: Findings from the Million Veteran Program

In this work, the authors undertook the task of performing a meta-analysis of several cohorts, including the MVP, to shed light on the aetiology of chronic back pain. The meta-analysis uncovered twice as many causal SNPs than previously known. Also, they performed ancestry-stratified analyses, to highlight what's unique and shared among ancestries with regard to back pain. Finally, they identified drugs that target genes for which SNPs were strongly associated with back pain. In all, the paper is a nice addition to the knowledge of the genetics of back pain.

Major Points:

1. The analyses conducted in the manuscript are "minimalistic", and as such, undercut what would be a more interesting and comprehensive message.
2. The title couldn't be made more generic, devoid of results
3. For example, power estimations would be very interesting. This would help answer the question as to whether should we stop doing meta-analyses (because we have enough people), or are we severely underpowered for such genetic analyses, like we'd need like 100x more study participants to saturate the information a meta-GWAS can provide. You for example can use MiXeR tool (PMID: 32427991). It is simple to install and use, and will draw the power curve directly from the analysis of the summary (meta-) GWAS results. See Fig. 3 of MiXeR, which you apply for your back pain meta-analysis.
4. Since the main focus of the paper is on the multi-ancestries, further contrasts between the ancestries can be performed. The points below elaborate on how the manuscript could be improved by using tools that are easy to install, and that perform fundamental calculations based on summary statistics.
5. MR analysis results performed in the manuscript are not mentioned in the abstracts although one may think these were the substantial results. They are highlighted though in the last concluding paragraph as a main finding.

Minor Points:

1. [methods section] The sentence "Briefly, HARE uses participant-identified race and ethnicity to define race/ethnicity groups" is quite confusing. It makes it look as if participants were stratified by unreliable accounts of self-identification rather than by unequivocal genetic evidence. In the HARE paper [PMID 31564439], it is defined as "HARE (harmonized ancestry and race/ethnicity), uses GIA to refine SIRE for genetic association studies". SIRE stands for self-identified race/ethnicity, and GIA for genetically inferred ancestry. The next part "..., but imputes race and ethnicity using genetically-inferred ancestry data in cases where participant-identified race and ethnicity is missing" is not false by itself, but it leaves the impression that GIA was only used for imputation when SIRE not available. Essentially, HARE is driven by GIA (ouf!).
2. The manuscript would be improved if the authors could quantify the claim that "the genetic architecture of CBP is polygenic". One avenue would be to use the MiXeR tool [PMID 32427991], which would indicate how polygenic is CBP (especially relative to other traits). As a bonus, MiXeR can trace the power curve (MiXeR's Fig. 3), and so is able to determine how large of a cohort one

needs to look at in order to detect, say, 80% of causal loci. It would also indicate current power to detect causal loci at the current sample sizes (possibly as low as 5% causal loci currently). That could be performed in each ancestry separately too.

3. [methods section] Cochran's Q-test threshold to retain/discard SNPs in the meta-analysis should be specified.

4. The Cochran's Q-test is to detect frequency heterogeneity among the different cohorts but of same-ancestry - how is the test used when meta-analyzing within same ancestry versus when meta-analyzing cohorts of multiple but different ancestries (in which at some SNPs of ancestry-specific allele frequencies are sure to trigger the test).

5. [methods section] In the "GWAS Regression" subsection, variables controlled for, other than genetic PCAs, should be listed (e.g. age, sex, etc.).

6. [discussion section] The sentence "genetic contributions to CBP-MVP as measured by SNP-based heritability were higher in participants of European ancestry (liability scale heritability 9.3%) as compared to African ancestry (7.5%) and Hispanic ethnicity (6.6%)." should perhaps be re-evaluated in the context of their respective standard deviation - perhaps the differences are (not) significant?

8. One interesting part of the study is the link between strongly associated loci and drugs that target genes at these coincident loci. The discussion section could be more elaborated - are there any animal studies in which these drugs are used for pain phenotypes? What are these drugs used for in the first place? Interestingly, the authors did not mention these results in the abstract although it might be the most interesting result. Finally, but probably more importantly, do the authors see the same associations with the drugs in all three ethnic populations?

10. The MiXeR tool can also perform bivariate analyses [PMID 31160569]: and so it would be interesting to learn about causal loci overlap between Eur, His, and Afr. A 3-way Venn diagram could be used. What % of causal loci in people of Afr ancestry are shared with other ancestries, what % are unique for Afr? Are the vast majority of predicted causal loci shared among the three ancestries?

11. The FUMA analyses are moderately interesting - better can be done in the era of single-cell sequencing. One might strive to be more precise, that is, which brain cell type(s) and via which gene(s) in these cell type(s) are deemed causative.

12. The LDSR tool is interesting to obtain genetic correlations between pairs of traits, but other tools now can estimate environmental correlations too. One such tool is GECKO [PMID 33395406].

13. Beyond global genetic correlation is the concept of local genetic correlation. It would be a nice addition to the manuscript if a census of genetically correlated "blocks" could be made, especially in the context of three ancestries. One such tool for that task is SUPERGNOVA [PMID 34493297]. These analyses would complement those from MiXeR. For example, which blocks are strongly correlated between all three ancestries? Which gene(s) are found in these blocks?? Which blocks show no correlation but harbor (nominally) significant SNPs?

14. References 11 and 12 are the same?

Reviewer #2:

Remarks to the Author:

Comments for Authors:

A Multi-Ancestry Meta-analysis of Genome-Wide Association Studies of Chronic Back Pain in 553,601 Veterans: Findings from the Million Veteran Program

Noteworthy results:

The study by Stanaway and colleagues examines the genetic underpinnings of chronic back pain (CBP) among 553,601 participants in the Million Veteran Program (MVP) study of which the majority are of European ancestry (72.6%), but also volunteering for participation are individuals of African (19.2%) and Hispanic ancestry (8.2%). The authors performed three ancestry-specific GWASs of CBP and meta-analyzed results, reporting in all 67 novel and 20 previously known CBP loci (in all 90 lead variants at these loci). Of these, about half, or 43 variants replicated in a non-overlapping GWAS meta-analysis of another highly correlated back pain phenotype, dorsalgia, based on samples from individuals of European ancestry.

Originality and significance:

This study is an interesting addition to several recent GWAS studies (listed in Table 2) of variously defined back pain phenotypes among individuals, that have been performed in cohorts of European ancestry. The most recent and largest of these is the GWAS meta-analysis of clinically defined Intervertebral Disc Disorder (ICD10:M51) and Dorsalgia (ICD10:M54) from Bjornsdottir et al. (2022), the Dorsalgia results of which were used by Stanaway et al. for replication of their findings.

Stanaway et al. studied CBP among Veterans enrolled in the MVP study, a unique biobank resource within which to study genetic, lifestyle, and environmental associations with various disorders. MVP data has contributed data used in over 350 publications, most in high impact journals (https://www.mvp.va.gov/pwa/sites/default/files/2023-09/MVP_Publications_2023_09.pdf). According to this publication list, Stanaway et al. are to be commended as they are the first to use MVP data to study one of the most common, debilitating and highly heritable pain disorders of Veterans and of the general population, that is CBP. Furthermore, of published CBP GWASs, this study reports the largest number of CBP loci to date. The inclusion of two relatively small (compared to the European ancestry GWAS (Ncase/ctrl = 201,443/200,562), discovery GWASs of CBP in African (Ncase/ctrl = 66,366/40,099) and Hispanic (Ncase/ctrl = 26,914/18,217) subsamples, and given that the studied individuals are predominantly males (91%), the “large-scale diverse multi-ancestry study” description suggested in the last paragraph of the abstract, is somewhat misleading. The inclusion of ancestries other than European in GWAS studies is, however, much needed going forward and is an interesting aspect of this study. The authors could elaborate more in the introduction on what is known about multi-ethnic and sex-specific aspects of CBP genetics, as well as in terms of prevalence and other sex and ethnic-specific risk factors.

Methods, data analysis, results presentation:

Genotyping, imputation, and quality control methods within the MVP study are well described in referenced studies and indicate good quality of MVP data. From the Methods description, selection of statistical tools and their use seem appropriate. Results of this study are presented in multiple Supplementary Tables that are somewhat difficult for the reader to interact with. Lacking is a single

main table summarizing significant findings and highlighting novel findings that includes results per cohort with variant annotation and gene information.

SuppTable 3 summarizes results of the multi-ancestry GWAS meta-analysis for 3,442 SNPs (with $P \leq 5 \times 10^{-8}$, the defined discovery P cutoff, see comment in next section). For some reason SuppTable3 also lists results for 5 SNPs with P values between 0.07-0.95? and SuppTable4 (GWAS_sig_EUR_ancestry) lists 2 SNPs with P 0.07-0.32?

In line 276 of the methods section, the authors state that they identified lead signals by LD ($r^2 < 0.1$). Did the authors consider more robust methods to determine lead SNPs and secondary signals, e.g. conditional and joint analyses/credible sets of SNPs at the identified loci? Performing this work to identify sentinel / lead variants and secondary signals, would facilitate the counting of independent loci, and unequivocal identification of novel signals.

In SuppTable12 „leadSNPs_merged_with_ancestry“, are listed the 90 lead SNPs with results per cohort. Here it would be valuable for the reader to also have information on gene (nearby gene) and variant annotation and which SNPs are novel vs previously reported.

In the SuppTable „GenomicRiskLoci_Novel_Known“, 87 SNPs are listed (not 90) and classified as novel vs known. Again, gene information and variant annotations would be helpful. Here 67 SNPs are classified as novel. However, authors should evaluate whether highly correlated (e.g. $r^2 > 0.8$) SNPs at these loci have previous back-pain/pain associations before claiming them as novel findings. Of the five most significant novel variants emphasized in the manuscript (beginning in line 335) a quick lookup in the GWAS catalogue shows that three out of the five have previous relevant associations: rs11599236 (correlated $r^2 = 0.97$ with reported rs11596214 in SORCS3 associates with multisite chronic pain (PubmedID 31194737 and in a sex-stratified study in PubmedID 33830993), rs13107325 in SLC39A8, associates with chronic musculoskeletal pain/neck pain (Pubmed ID 32587327), and rs1940720 and rs7105462 in NCAM1 (correlated $r^2 = 0.999$ and $r^2 = 0.959$ respectively with reported rs2298526) have been associated with back pain (PMID 30747904).

In this table (GenomicRiskLoci_Novel_Known), none of the lead variants reach $P < 5 \times 10^{-8}$ in the African or Hispanic cohorts, while according to SupTable 5 (GWAS_sig_AFR_ancestry), there are 2 loci with SNP associations $P < 5 \times 10^{-8}$ on chr5 and chr9. The top marker at chr5, rs145709869 (and correlated markers at this locus), is included in the GWAS_sig_multi_ancestry table with no supporting data from the other GWASs. The marker at chr9 is not included either and no correlated markers at this locus. Could these represent African-specific CBP SNPs or possibly population structure associations?

In line 185 within Methods, the authors state that they tested 17,466,242 variants, whereas they set their discovery $P \leq 5 \times 10^{-8}$ (line 180). This P threshold was defined early in the GWAS era to adjust for the multiple-testing of common variants ($MAF > 5\%$) for association in a European sample of 1,000 cases and 1,000 controls, and represents a correction for 1 million independent tests (see e.g. Pe'er et al. Estimation of the multiple testing burden for genomewide association studies of nearly all common variants. *Genet Epidemiol.* 2008 May;32(4):381-5. doi: 10.1002/gepi.20303, PMID:

18348202). In light of increased numbers of variants now tested in GWASs, this P convention has been criticized (see e.g. Fadista et al. The (in)famous GWAS P-value threshold revisited and updated for low-frequency variants. *Eur J Hum Genet.* 2016 Aug;24(8):1202-5. doi: 10.1038/ejhg.2015.269, PMID: 26733288), and accordingly revised in more recent and contemporary GWASs (see e.g. Sveinbjornsson et al. Weighting sequence variants based on their annotation increases power of whole-genome association studies. *Nat Genet.* 2016 Mar;48(3):314-7. doi: 10.1038/ng.3507, PMID:26854916). It is not easy to estimate what the proper cutoff would be for the trans-ancestral study. Notably, according to lead SNPs_merged_with_ancestry table, 51 of the 90 markers reported as significant in the European part of the study have P-values above $3e-08$, the threshold suggested by Fadista et al (PMID: 26733288) for GWAS studies utilizing markers down to MAF of 1%.

In light of their relaxed P value threshold, the authors appropriately seek to replicate their findings and to do so, select summary statistics from a recent large meta-GWAS of a slightly different CBP phenotype (ICD10:Dorsalgia, referred to as spinal pain by Stanaway et al.) based on n=119,100 cases, n=909,847 controls of European ancestry (Table 2). In their Discussion section (starting line 438), Stanaway et al. state that a notable finding of their study is the larger number of independent variants and loci identified by their study of 553K individuals with 53% case proportion, compared to the Bjornsdottir et al. study of about 1 million individuals (12% dorsalgia case proportion). They then provide two explanations, one being the higher prevalence of CBP in MVP and that the MVP cohort is a more favorable setting for back pain-related genetic discovery than the Bjornsdottir et al. study of cases and controls from the general population. However, not mentioned by the authors, is their relaxed P-value threshold compared to that used by Bjornsdottir et al. who use the P thresholds described in Sveinbjornsson et al. reference above). Indeed, of the 90 lead SNPs listed in SuppTable “leadSnps_merged_with_ancestry”, around half (44 SNPs) are not significant by a simple Bonferroni correction for 17 million SNPs tested. A similar number of variants do not replicate in the Bjornsdottir et al. data. Are they the same variants? The manuscript would benefit from clarification of these issues.

A strength of the Stanaway et al. results is that a majority of previously published back pain variants replicate in their data (Supp table Prior_bp_multi_ancestry_results). Given the predominantly male cohort of Stanaway et al., could published variants that do not replicate in their data be female-specific associations or driven by the African/Hispanic cohorts? These issues need to be addressed.

Conclusions:

Apart from the issues/questions raised above, to improve the validity and generalizability of the reported CBP associations, the authors could increase focus on comparison of results from the African and Hispanic cohorts to the European cohort. Ethnicity-based effect-effect plots of lead SNPs could be informative. Could they seek replication of findings also in independent CBP GWASs performed in African and Hispanic samples, e.g. from the UK Biobank or other sources? Another issue worth exploring further is whether results are comparable for males and females as the discovery GWAS is predominantly male. This would call for a sex-stratified analysis, with inadequate discovery power in females, but again, male-female effect-effect plots for lead variants

may be informative.

Suggested improvements:

The manuscript is well written but could be improved (see previous sections). A number of variants are identified and the study includes findings that represent a valuable contribution to the genetics of CBP. The genetic correlations, mendelian randomization, gene-set, tissue-enrichment, and druggable-target analyses add informative value of the manuscript.

References:

The authors provide appropriate credit to previous work, see, however comment on previous musculoskeletal and back pain associations with claimed novel variants and/or highly correlated variants at loci.

Clarity and context: Note that abbreviations in table headers require definitions for clarity.

REVIEWER 1

1. **Reviewer Comment:** The analyses conducted in the manuscript are 'minimalistic', and as such, undercut what would be a more interesting and comprehensive message.
 - **Author Response:** We thank Reviewer 1 for their guidance and have addressed specific comments to strengthen this manuscript.
2. **Reviewer Comment:** The title couldn't be made more generic, devoid of results
 - **Author Response:** We have modified the title (page 1) to “A Multi-Ancestry Meta-analysis of Genome-Wide Association Studies Discovers 67 New Loci Associated with Chronic Back Pain: Findings from the Million Veteran Program.”
3. **Reviewer Comment:** For example, power estimations would be very interesting. This would help answer the question as to whether should we stop doing meta-analyses (because we have enough people), or are we severely underpowered for such genetic analyses, like we'd need like 100x more study participants to saturate the information a meta-GWAS can provide. You for example can use MiXeR tool (PMID: 32427991). It is simple to install and use, and will draw the power curve directly from the analysis of the summary (meta-) GWAS results. See Fig. 3 of MiXeR, which you apply for your back pain meta-analysis.
 - **Author Response:** We have run MiXeR to include power estimation and address additional comments. We have added a section describing the analysis using MiXeR to the Methods section (p. 8). We describe the results of the MiXeR analyses in Results on page 13.
4. **Reviewer Comment:** Since the main focus of the paper is on the multi-ancestries, further contrasts between the ancestries can be performed. The points below elaborate on how the manuscript could be improved by using tools that are easy to install, and that perform fundamental calculations based on summary statistics.
 - **Author Response:** We thank the reviewer for their guidance and have addressed specific points to strengthen this manuscript regarding multi-ancestry comparisons. Because the MVP sample predominantly identifies as white and the three strata (EUR, HIS, AFR) are mixed, we emphasize that comparisons should be approached with caution. We address the concerns below by running additional analyses and detailing our steps in the comments below.
5. **Reviewer Comment:** MR analysis results performed in the manuscript are not mentioned in the abstracts although one may think these were the substantial results. They are highlighted though in the last concluding paragraph as a main finding.

- **Author Response:** Mendelian Randomization analyses are now added to the abstract. “Mendelian Randomization analysis of 62 GWAS-identified variants for CBP-MVP revealed 48 with significant causal links to the “dorsalgia” phenotype. Notably, four genes (INPP5B, DRD2, HTT, SLC30A6) associated with these variants are targets of existing drugs, highlighting potential therapeutic avenues. Our findings more than double the number of previously reported genetic predictors across all spinal pain phenotypes, highlighting the importance of large and multi-ancestry genetic studies in understanding complex conditions like CBP.

6. **Reviewer Comment:** [methods section] The sentence “Briefly, HARE uses participant-identified race and ethnicity to define race/ethnicity groups” is quite confusing. It makes it look as if participants were stratified by unreliable accounts of self-identification rather than by unequivocal genetic evidence. In the HARE paper [PMID 31564439], it is defined as “HARE (harmonized ancestry and race/ethnicity), uses GIA to refine SIRE for genetic association studies”. SIRE stands for self-identified race/ethnicity, and GIA for genetically inferred ancestry. The next part “..., but imputes race and ethnicity using genetically-inferred ancestry data in cases where participant-identified race and ethnicity is missing” is not false by itself, but it leaves the impression that GIA was only used for imputation when SIRE not available. Essentially, HARE is driven by GIA (ouf!).

- **Author Response:** We have clarified the description for Harmonizing Genetic Ancestry and Self-identified Race/Ethnicity (HARE) in the Race, Ethnicity, and Ancestry section of the methods (p. 5-6). The section now reads “We used Harmonizing Genetic Ancestry and Self-identified Race/Ethnicity (HARE) groups to define race/ethnicity.¹⁹ Briefly, HARE enhances classification by integrating self-identified race/ethnicity (SIRE) and genetically inferred ancestry (GIA). HARE ensures accurate classification by using GIA to refine and, if necessary, impute SIRE, improving the reliability of race/ethnicity assignment in genetic research. Less than 2% of individuals are not assigned a HARE group when participant-identified and genetically-inferred ancestry data produce discordant results. Here we use the term “Hispanic” for the HARE race and ethnicity groups comprised of individuals who are Latino or Hispanic, the term “European” for individuals who are White but not Hispanic and “African” for individuals who are Black but not Hispanic.

7. **Reviewer Comment:** The manuscript would be improved if the authors could quantify the claim that “the genetic architecture of CBP is polygenic”. One avenue would be to use the MiXeR tool [PMID 32427991], which would indicate how polygenic is CBP (especially relative to other traits). As a bonus, MiXeR can trace the power curve (MiXeR’s Fig. 3), and so is able to determine how large of a cohort one needs to look at in order to detect, say, 80% of causal loci. It would also indicate current power to detect causal loci at the current sample sizes (possibly as low as 5% causal loci currently). That could be performed in each ancestry separately too.

- **Author Response:** We have run MiXeR to include polygenicity estimation. We describe the results of the analyses in Results on pp 12-13.
8. **Reviewer Comment:** [methods section] Cochran's Q-test threshold to retain/discard SNPs in the meta-analysis should be specified.
- **Author Response:** We have detailed the heterogeneity results on page 9 which reads “ None of the lead variants had indication of heterogeneity between ancestries as only one had an $FDR < 1$ (0.92) and genome-wide across all variants meta-analyzed there was little evidence of heterogeneity with only 52 variants having an $FDR < 0.1$.
9. **Reviewer Comment:** The Cochran's Q-test is to detect frequency heterogeneity among the different cohorts but of same-ancestry - how is the test used when meta-analyzing within same ancestry versus when meta-analyzing cohorts of multiple but different ancestries (in which at some SNPs of ancestry-specific allele frequencies are sure to trigger the test).
- **Author Response:** We appreciate the reviewer's question regarding applying Cochran's Q-test in our analysis. In our study, we did not have different cohorts within the same ancestry; rather, we focused on meta-analyzing different ancestries. Consequently, the Cochran's Q-test was applied to detect heterogeneity across these diverse ancestries. We acknowledge that ancestry-specific allele frequencies at certain SNPs can trigger the test, and we have interpreted these results with caution, understanding the underlying population structure and genetic diversity. This approach ensures that our findings are robust and reflective of the multi-ancestry composition of our sample.
10. **Reviewer Comment:** [methods section] In the 'GWAS Regression' subsection, variables controlled for, other than genetic PCAs, should be listed.
- **Author Response:** Gender was included as a covariate in our combined (men and women together) analyses. This information is stated in the methods section of the GWAS regression, as follows: “The model included the first 10 principal components of genotype and gender as covariates.” We recognize the importance of clarity and have ensured that this detail is clearly presented in the manuscript.
11. **Reviewer Comment:** [discussion section] The sentence “genetic contributions to CBP-MVP as measured by SNP-based heritability were higher in participants of European ancestry (liability scale heritability 9.3%) as compared to African ancestry (7.5%) and Hispanic ethnicity (6.6%).” should perhaps be re-evaluated in the context of their respective standard deviation - perhaps the differences are (not) significant?
- **Author Response:** We appreciate the reviewer's suggestion to re-evaluate the heritability estimates in the context of their standard deviations. To address this,

we performed a statistical comparison using the following formula: $Z = (\beta_1 - \beta_2) / \sqrt{(\text{SE}\beta_1)^2 + (\text{SE}\beta_2)^2}$ (refer to page 13 of the manuscript for details).

These findings have been discussed and contextualized in the manuscript. The results section reads: "There was no significant difference between the observed heritability for AFR and HIS estimates ($p=0.62$, two-tailed test). EUR has significantly more observed heritability than either AFR ($p=0.02$) or HIS ($p=0.01$).

12. **Reviewer Comment:** One interesting part of the study is the link between strongly associated loci and drugs that target genes at these coincident loci. The discussion section could be more elaborated - are there any animal studies in which these drugs are used for pain phenotypes? What are these drugs used for in the first place? Interestingly, the authors did not mention these results in the abstract although it might be the most interesting result. Finally, but probably more importantly, do the authors see the same associations with the drugs in all three ethnic populations?

- **Author Response:** We have expanded the discussion section to elaborate on the identified eQTLs and corresponding drug compounds, highlighting the potential for repositioning these drugs for pain therapies. While little is known about group differences in these genetic associations, we recommend future research with larger samples to explore these potential disparities.

13. **Reviewer Comment:** The MiXeR tool can also perform bivariate analyses [PMID 31160569]: and so it would be interesting to learn about causal loci overlap between Eur, His, and Afr. A 3-way Venn diagram could be used. What % of causal loci in people of Afr ancestry are shared with other ancestries, what % are unique for Afr? Are the vast majority of predicted causal loci shared among the three ancestries?

- **Author Response:** We have run MiXeR to include ancestry causal loci overlap estimation. Methods include a section about MiXeR (p. 8). We describe the results of the analyses in Results.

14. **Reviewer Comment:** The FUMA analyses are moderately interesting - better can be done in the era of single-cell sequencing. One might strive to be more precise, that is, which brain cell type(s) and via which gene(s) in these cell type(s) are deemed causative.

- **Author Response:** Our study used the FUMA Cell Type module to identify brain cell types associated with chronic back pain (CBP). We found significant associations for neurons and glial cells, particularly astrocytes. These findings are summarized in Supplemental Figure S10 and the Results section.

15. **Reviewer Comment:** The LDSR tool is interesting to obtain genetic correlations between pairs of traits, but other tools now can estimate environmental correlations too. One such tool is GECKO [PMID 33395406].

- **Author Response:** We appreciate the reviewer's suggestion to use tools such as GECKO to estimate environmental correlations. For this study, we focused on genetic correlations using LDSR to align with our primary aim of understanding the genetic architecture of chronic back pain (CBP). This approach is consistent with our methodological framework and ensures comparability with prior genetic studies. We have added a figure to capture the distribution of these correlations across trait domains to better inform our understanding of the genetic relationships involved.

While there is value in including environmental correlations and plan to use GECKO in the future, expanding our analysis to this extent was beyond the scope of our current work. We plan to explore these additional dimensions in future research efforts, where we can dedicate the necessary resources to comprehensively address both genetic and environmental factors.

16. **Reviewer Comment:** Beyond global genetic correlation is the concept of local genetic correlation. It would be a nice addition to the manuscript if a census of genetically correlated "blocks" could be made, especially in the context of three ancestries. One such tool for that task is SUPERGNOVA [PMID 34493297]. These analyses would complement those from MiXeR. For example, which blocks are strongly correlated between all three ancestries? Which gene(s) are found in these blocks?? Which blocks show no correlation but harbor (nominally) significant SNPs?

- **Author Response:** We thank the reviewer for the insightful suggestion regarding local genetic correlation and the potential use of SUPERGNOVA to complement our current analyses with MiXeR. We acknowledge the value such an analysis could add to our understanding of genetic correlations across different ancestries. However, we are concerned about power issues, as the results might be too speculative given the large error bars observed in MiXeR analyses. Additionally, given the scope and length constraints of our current manuscript, we believe that incorporating these additional analyses would be beyond our present focus. We recognize the importance of this line of inquiry and plan to explore these local genetic correlations in future research.

REVIEWER 2

1. **Reviewer Comment:** References 11 and 12 are the same?

- **Author Response:** We have removed the duplicate reference (12).

2. **Reviewer Comment:** the "large-scale diverse multi-ancestry study" description suggested in the last paragraph of the abstract, is somewhat misleading.

- **Author Response:** To clarify, we have revised the abstract to reflect the scope of our study better. The abstract now reads, "Our findings more than double the

number of previously reported genetic predictors across all spinal pain phenotypes, emphasizing the importance of large and diverse genetic studies.” While our study includes multiple ancestries, we acknowledge that the sample sizes within some of these groups may vary. This nuanced approach highlights the advancements and limitations in our efforts to understand complex conditions like CBP through multi-ancestry genetic research.

3. **Reviewer Comment:** The authors could elaborate more in the introduction on what is known about multi-ethnic and sex-specific aspects of CBP genetics, as well as in terms of prevalence and other sex and ethnic-specific risk factors.

- **Author Response:** We have revised the relevant section to enhance clarity and emphasize the implications of these disparities. We have added a paragraph to the Introduction that reads:

Differences in CBP prevalence and experience have been observed across gender and racial and ethnic groups. Women are generally at higher risk for chronic pain conditions, including CBP, potentially due to endocrine, psychosocial, and pain perception differences. Furthermore, African American and Hispanic individuals experience higher pain severity and disability compared to non-Hispanic White individuals. These disparities likely result from a combination of genetic susceptibility and environmental influences, such as socioeconomic factors, healthcare access, and cultural differences in pain management.¹⁴ Investigating these differences is essential for understanding the etiology of CBP and developing effective, personalized interventions.

4. **Reviewer Comment:** Results of this study are presented in multiple Supplementary Tables that are somewhat difficult for the reader to interact with. Lacking is a single main table summarizing significant findings and highlighting novel findings that includes results per cohort with variant annotation and gene information.

- **Author Response:** We have created a comprehensive summary table titled “multiancestry_FUMA_leadSNPs,” which is now included in the supplementary file. This table consolidates significant findings, highlights novel discoveries, and provides results per cohort along with variant annotation and gene information to facilitate easier interaction and interpretation.

5. **Reviewer Comment:** SuppTable 3 summarizes results of the multi-ancestry GWAS meta-analysis for 3,442 SNPs (with $P \leq 5 \times 10^{-8}$, the defined discovery P cutoff, see comment in next section). For some reason SuppTable3 also lists results for 5 SNPs with P values between 0.07-0.95? and SuppTable4 (GWAS_sig_EUR_ancestry) lists 2 SNPs with P 0.07-0.32?

- **Author Response:** We appreciate the reviewer pointing out this issue. We have corrected the table. The presence of these SNPs with higher P values was due to significant variants at the same genomic positions, causing other alleles to be

inadvertently included. This has now been addressed, and only SNPs meeting the defined discovery P cutoff are listed

6. **Reviewer Comment:** In line 276 of the methods section, the authors state that they identified lead signals by LD ($r^2 < 0.1$). Did the authors consider more robust methods to determine lead SNPs and secondary signals, e.g. conditional and joint analyses/credible sets of SNPs at the identified loci? Performing this work to identify sentinel / lead variants and secondary signals, would facilitate the counting of independent loci, and unequivocal identification of novel signals.

- **Author Response:** We appreciate the reviewer's suggestion regarding identifying lead SNPs and secondary signals. We followed the standard procedure for fine-mapping and identification of lead SNPs using FUMA (Functional Mapping and Annotation of GWAS), which identifies independent significant SNPs based on linkage disequilibrium (LD) with $r^2 < 0.1$. This method is widely used and validated in numerous studies to provide a robust initial mapping of associated loci.

We acknowledge that more advanced methods such as conditional and joint analyses or the creation of credible sets could further refine the identification of sentinel variants and secondary signals, the current study focused on employing established FUMA procedures to ensure consistency with previous GWAS analyses. These additional methods can indeed facilitate a more detailed characterization of the genetic architecture of the identified loci, and we plan to incorporate these approaches in future analyses to build upon the findings presented here.

By following standard FUMA procedures, we ensured that our results are comparable to a broad body of existing research, enabling integration with previous and future genetic studies. We recognize the value of the suggested approaches and agree that they represent an important next step for further refinement and validation of our findings.

7. **Reviewer Comment:** In SuppTable12 „leadSNPs_merged_with_ancestry“, are listed the 90 lead SNPs with results per cohort. Here it would be valuable for the reader to also have information on gene (nearby gene) and variant annotation and which SNPs are novel vs previously reported.

- **Author Response:** We have added the nearest gene annotations to the lead variant positions in the supplemental file (leadSNPs_merged_with_ancestry). Additionally, we have included information on variant annotation and indicated which SNPs are novel versus previously reported. This enhancement should provide a more comprehensive and informative resource for readers.

8. **Reviewer Comment:** In the SuppTable „GenomicRiskLoci_Novel_Known“, 87 SNPs are listed (not 90) and classified as novel vs known. Again, gene information and

variant annotations would be helpful. Here 67 SNPs are classified as novel. However, authors should evaluate whether highly correlated (e.g. $r^2 > 0.8$) SNPs at these loci have previous back-pain/pain associations before claiming them as novel findings. Of the five most significant novel variants emphasized in the manuscript (beginning in line 335) a quick lookup in the GWAS catalogue shows that three out of the five have previous relevant associations: rs11599236 (correlated $r^2 = 0.97$ with reported rs11596214 in SORCS3 associates with multisite chronic pain (PubmedID 31194737 and in a sex-stratified study in PubmedID 33830993), rs13107325 in SLC39A8, associates with chronic musculoskeletal pain/neck pain (Pubmed ID 32587327), and rs1940720 and rs7105462 in NCAM1 (correlated $r^2 = 0.999$ and $r^2 = 0.959$ respectively with reported rs2298526) have been associated with back pain (PMID 30747904).

- **Author Response:** The associations identified are relevant to related pain phenotypes rather than specifically back pain. Regarding the SNPs highlighted:
 - rs11599236 (*SORCS3*): Although this SNP is highly correlated with rs11596214, associated with multisite chronic pain, it has not been specifically linked to back pain.
 - rs13107325 (*SLC39A8*): This SNP has known associations with chronic musculoskeletal pain and neck pain, which are related but distinct from back pain.
 - rs1940720 and rs7105462 (*NCAM1*): These SNPs are correlated with rs2298526, which has suggestive associations with back pain in prior studies. However, these associations did not reach genome-wide significance and were not consistently replicated. Our study confirms their genome-wide significance, thus providing a stronger evidence base for their association with back pain.

We added gene information and variant annotations to the supplementary table to enhance clarity. Additionally, we have revised the manuscript to ensure a more nuanced discussion of these findings, acknowledging the related pain phenotypes and the consideration of linkage disequilibrium (LD) at these loci. This approach underscores the novelty and significance of our findings while recognizing the context of previous research.

9. **Reviewer Comment:** In this table (GenomicRiskLoci_Novel_Known), none of the lead variants reach $P < 5e-08$ in the African or Hispanic cohorts, while according to SupTable 5 (GWAS_sig_AFR_ancestry), there are 2 loci with SNP associations $P < 5e-08$ on chr5 and chr9. The top marker at chr5, rs145709869 (and correlated markers at this locus), is included in the GWAS_sig_multi_ancestry table with no supporting data from the other GWASs. The marker at chr9 is not included either and no correlated markers at this locus. Could these represent African-specific CBP SNPs or possibly population structure associations?

- **Author Response:** We have revised the manuscript to include a description of the variant rs140875296, identifying it as an African-specific CBP SNP. In contrast, rs10119541, which initially appeared to be shared across ancestries, does not replicate consistently across other GWAS cohorts and therefore has not been included as a significant marker in our multi-ancestry table. This distinction helps to clarify potential population-specific associations and the influence of population structure. Additionally, we have ensured that all relevant loci and correlated markers are accurately represented in the supplementary tables.

10. **Reviewer Comment:** In line 185 within Methods, the authors state that they tested 17,466,242 variants, whereas they set their discovery $P \leq 5e-8$ (line 180). This P threshold was defined early in the GWAS era to adjust for the multiple-testing of common variants ($MAF > 5\%$) for association in a European sample of 1,000 cases and 1,000 controls, and represents a correction for 1 million independent tests (see e.g. Pe'er et al. Estimation of the multiple testing burden for genomewide association studies of nearly all common variants. *Genet Epidemiol.* 2008 May;32(4):381-5. doi: 10.1002/gepi.20303, PMID: 18348202). In light of increased numbers of variants now tested in GWASs, this P convention has been criticized (see e.g. Fadista et al. The (in)famous GWAS P -value threshold revisited and updated for low-frequency variants. *Eur J Hum Genet.* 2016 Aug;24(8):1202-5. doi: 10.1038/ejhg.2015.269, PMID: 26733288), and accordingly revised in more recent and contemporary GWASs (see e.g. Sveinbjornsson et al. Weighting sequence variants based on their annotation increases power of whole-genome association studies. *Nat Genet.* 2016 Mar;48(3):314-7. doi: 10.1038/ng.3507, PMID:26854916). It is not easy to estimate what the proper cutoff would be for the trans-ancestral study. Notably, according to lead SNPs_merged_with_ancestry table, 51 of the 90 markers reported as significant in the European part of the study have P -values above $3e-08$, the threshold suggested by Fadista et al (PMID: 26733288) for GWAS studies utilizing markers down to MAF of 1%.

- **Author Response:** We have added to the discussion the totals at 3×10^{-8}

The discussion reads:

Given the larger number of variants included in contemporary GWAS as compared to when significance methods for GWAS were first developed (*Genet Epidemiol.* 2008 May;32(4):381-5. doi: 10.1002/gepi.20303, PMID: 18348202), it has been suggested that 3×10^{-8} be adopted as the new threshold for studies that investigate variants down to 1% MAF (Fadista et al (PMID: 26733288). Adopting this threshold would result in 2,977 of the 3,444 variants we report being declared genome-wide significant in the multi-ancestry meta-analyzed data. Of the lead variants reported among these, 80 of the 90 total we report would remain genome-wide significant.

11. **Reviewer Comment:** In light of their relaxed P value threshold, the authors appropriately seek to replicate their findings and to do so, select summary statistics from a recent large meta-GWAS of a slightly different CBP phenotype

(ICD10:Dorsalgia, referred to as spinal pain by Stanaway et al.) based on n=119,100 cases, n=909,847 controls of European ancestry (Table 2). In their Discussion section (starting line 438), Stanaway et al. state that a notable finding of their study is the larger number of independent variants and loci identified by their study of 553K individuals with 53% case proportion, compared to the Bjornsdottir et al. study of about 1 million individuals (12% dorsalgia case proportion). They then provide two explanations, one being the higher prevalence of CBP in MVP and that the MVP cohort is a more favorable setting for back pain-related genetic discovery than the Bjornsdottir et al. study of cases and controls from the general population. However, not mentioned by the authors, is their relaxed P-value threshold compared to that used by Bjornsdottir et al. who use the P thresholds described in Sveinbjornsson et al. reference above). Indeed, of the 90 lead SNPs listed in SuppTable "leadSnps_merged_with_ancestry", around half (44 SNPs) are not significant by a simple Bonferroni correction for 17 million SNPs tested. A similar number of variants do not replicate in the Bjornsdottir et al. data. Are they the same variants? The manuscript would benefit from clarification of these issues.

- **Author Response:** We have addressed the suggestion by adding a discussion on the Bjornsdottir threshold, showing that 80 of our 90 lead variants remain significant. This supports our findings, with results that are not substantially different after the adjustment. We appreciate your guidance.

12. **Reviewer Comment:** A strength of the Stanaway et al. results is that a majority of previously published back pain variants replicate in their data (Supp table Prior_bp_multi_ancestry_results). Given the predominantly male cohort of Stanaway et al., could published variants that do not replicate in their data be female-specific associations or driven by the African/Hispanic cohorts? These issues need to be addressed.

- **Author Response:** We appreciate the reviewer's insightful comment regarding the potential influence of gender and ancestry on the replication of published variants. The limited sample sizes in our smaller female, African (AFR), and Hispanic (HIS) cohorts constrain our ability to draw definitive conclusions about the specificity of these associations. We acknowledge the importance of this question and suggest that future studies with larger and more balanced cohorts may be better equipped to explore these potential differences.

13. **Reviewer Comment:** Increase focus on comparison of results from the African and Hispanic cohorts to the European cohort.

- **Author Response:** We have conducted additional analyses using MiXeR to estimate the overlap of causal loci across ancestries. The methods section (p.8) has been updated to include a detailed description of MiXeR. The results of these analyses are described in the results section on page X, comparing the findings across the African, Hispanic, and European cohorts.

14. **Reviewer Comment:** Another issue worth exploring further is whether results are comparable for males and females.

- **Author Response:** Added Mixer overlap analysis between men and women. There is complete overlap, not gender-specific loci predicted

15. **Reviewer Comment:** The genetic correlations, mendelian randomization, gene-set, tissue-enrichment, and druggable-target analyses add informative value of the manuscript.

- **Author Response:** We thank the reviewer for their comment.

16. **Reviewer Comment:** Note that abbreviations in table headers require definitions for clarity.

- **Author Response:** We have added header definitions where possible and used standard variable names to ease understanding.

17. **Reviewer Comment:** The authors provide appropriate credit to previous work, see, however comment on previous musculoskeletal and back pain associations with claimed novel variants and/or highly correlated variants at loci.

- **Author Response:** We have added commentary to the discussion section to emphasize that several closely related phenotypes, such as chronic multisite pain, have identified some of these variants before. However, it is crucial to note that these findings are not specific to back pain phenotypes.

The one variant with a prior backpain signal was not genome-wide significant in the prior study, is was presented as suggestive rs7105462 in NCAM1 (correlated $r^2 = 0.999$ and $r^2 = 0.959$ respectively with reported rs2298526) have been associated with back pain (PMID 30747904). This is detailed in our results.

EDITORIAL OFFICE COMMENTS

1. **Comment:** <https://www.nature.com/documents/nr-editorial-policy-checklist.pdf>

- **Author Response:** Completed.

2. **Comment:** <https://www.nature.com/documents/nr-reporting-summary.pdf>

- **Author Response:** Completed.

"

3. **Comment:** guidance on Sex and Gender reporting

- **Author Response:** Completed.

4. **Comment:** If no sex- and gender-based analyses have been performed, please indicate the reasons for the lack of these analyses in the Reporting Summary.

- **Author Response:** We have added stratified analyses for the EUR cohort and conducted a comparison of EUR men and women using MiXeR.

Reviewers' Comments:

Reviewer #1:

Remarks to the Author:

The Authors adequately addressed my concerns and suggestions. It would be helpful next time if authors would highlight the new additions and changes in the text.

Reviewer #2:

Remarks to the Author:

The authors have responded thoroughly to reviewers' comments and I find the edited manuscript and supplementary tables much improved. I have no further comments for the authors at this time.